# MiCloud: A unified web platform for comprehensive microbiome data analysis

**Won Gu**[1◦], **Jeongsup Moon**[2◦], **Crispen Chisina**[1], **Byungkon Kang**[3], **Taesung Park**[2,4‡]*, **Hyunwook Koh**[1‡]*

**1** Department of Applied Mathematics and Statistics, The State University of New York, Korea, Incheon, South Korea, **2** Interdisciplinary Program in Bioinformatics, Seoul National University, Seoul, South Korea, **3** Department of Computer Science, The State University of New York, Korea, Incheon, South Korea, **4** Department of Statistics, Seoul National University, Seoul, South Korea

◦ These authors contributed equally to this work.
‡ TP and HK also contributed equally to this work.
* tspark@stats.snu.ac.kr (TP); hyunwook.koh@stonybrook.edu (HK)

**Data Availability Statement:** The raw sequence data for the UK twin study are publicly available in the European Bioinformatics Institute (EMBL-EBI) database (access number: ERP006339 and ERP006342) (Goodrich et al., 2014). The feature

## Abstract

The recent advance in massively parallel sequencing has enabled accurate microbiome profiling at a dramatically lowered cost. Then, the human microbiome has been the subject of intensive investigation in public health and medicine. In the meanwhile, researchers have developed lots of microbiome data analysis methods, protocols, and/or tools. Among those, especially, the web platforms can be highlighted because of the user-friendly interfaces and streamlined protocols for a long sequence of analytic procedures. However, existing web platforms can handle only a categorical trait of interest, cross-sectional study design, and the analysis with no covariate adjustment. We therefore introduce here a unified web platform, named MiCloud, for a binary or continuous trait of interest, cross-sectional or longitudinal/family-based study design, and with or without covariate adjustment. MiCloud handles all such types of analyses for both ecological measures (i.e., alpha and beta diversity indices) and microbial taxa in relative abundance on different taxonomic levels (i.e., phylum, class, order, family, genus and species). Importantly, MiCloud also provides a unified analytic protocol that streamlines data inputs, quality controls, data transformations, statistical methods and visualizations with vastly extended utility and flexibility that are suited to microbiome data analysis. We illustrate the use of MiCloud through the United Kingdom twin study on the association between gut microbiome and body mass index adjusting for age. MiCloud can be implemented on either the web server (http://micloud.kr) or the user's computer (https://github.com/wg99526/micloudgit).

## Introduction

The human microbiome is the entire set of all microbes that live in and on the human body. The recent advance in massively parallel sequencing has enabled accurate microbiome profiling at a dramatically lowered cost. Then, the human microbiome has been the subject of intensive investigation in public health and medicine. Researchers have, for example, found numerous microbiome-associated disorders (e.g., obesity [1, 2], diabetes [3, 4], inflammatory

table, taxonomic table, phylogenetic tree and metadata for the UK twin study are publicly available as example 16S data on MiCloud. The URLs for MiCloud are http://micloud.kr (web application) and https://github.com/wg99526/micloudgit (GitHub).

**Funding:** HK was supported by the National Research Foundation of Korea (NRF) grant funded by the Korean government (MSIT) (2021R1C1C1013861) and by Incheon Technopark. TP was supported by the Bio & Medical Technology Development Program of the National Research Foundation of Korea (NRF) grant (2013M3A9C4078158).

**Competing interests:** The authors have declared that no competing interests exist.

bowel disease [5], cancers [6–11]), behavioral/environmental factors (e.g., diet [12], residence [13], smoking [14]), medical interventions (e.g., antibiotics [3], non-antibiotic drugs [15]), and so forth.

In the meanwhile, researchers have also developed lots of microbiome data analysis methods, protocols and/or tools. For example, microbiome profiling has been streamlined by the recent bioinformatic pipelines, such as QIIME [16], MG-RAST [17], Mothur [18], MEGAN [19] and MetaPhlAn [20]. Researchers can thereby easily process raw sequence data from either 16S rRNA amplicon sequencing [16, 21] or shotgun metagenomics [22], and acquire precise metagenomic information on microbial abundance, taxonomic annotation, gene/functional attribute and phylogenetic tree [23]. A variety of downstream data analysis methods have also been developed for ecological (e.g., PERMANOVA [24–26], MiRKAT [27, 28], aMiAD [29]), taxonomical (e.g., metagenomeSeq [30], ANCOM [31]) and functional (e.g., STAMP [32]) analysis, and their software packages are widely available.

We especially note here that recent web platforms have empowered user-friendly and interactive operations over the past command-line analytic tools. Besides, the web platforms provide standardized protocols for a long sequence of analytic procedures in data filtering, quality control, data transformation and analysis. Hence, even non-professional programmers like clinicians and biologists can easily deal with the microbiome data, and it is straightforward to reproduce the results. However, existing web platforms for downstream microbiome data analysis, including MicrobiomeAnalyst [33], METAGENassist [34] and EzBioCloud [35], can handle only a categorical trait of interest (e.g., diseased vs. healthy, treatment vs. placebo), cross-sectional study, and the analysis with no covariate adjustment. Yet, in microbiome studies, researchers often employ family-based or longitudinal study designs [2, 3] to survey different types of traits. Especially, in observational studies, the covariate-adjusted analyses are necessary to properly control for potential confounders (e.g., age, gender).

We introduce here a unified web platform, named MiCloud, for comprehensive microbiome data analysis. MiCloud performs microbiome data analysis for a binary (e.g., diseased vs. healthy, treatment vs. placebo) or continuous (e.g., body mass index, immune/metabolic activity level, brain quotient) trait of interest, cross-sectional or longitudinal/family-based study design, and with or without covariate adjustment with respect to both ecological measures (i.e., alpha and beta diversity indices) and microbial taxa in relative abundance on different taxonomic levels (i.e., phylum, class, order, family, genus and species). Moreover, MiCloud provides a unified analytic protocol that streamlines data inputs (individual and integrated data forms), quality controls (with respect to kingdom, library size, mean proportion and taxonomic name), data transformations (various alpha and beta diversity indices, and taxonomic abundance forms of count, rarefied count [36], proportion and centered log-ratio (CLR) [37]), statistical methods (various methods for different study designs, data forms and analytic schemes) and visualizations (various plots for data summary and ecological/taxonomical analyses) that are suited to microbiome data analysis. Therefore, users can enjoy comprehensive microbiome data analysis on user-friendly web environments with vastly extended utility and flexibility. MiCloud can be implemented on the web server or locally on the user's computer when the web server is busy.

The rest of the paper is organized as follows. In *Materials and Methods*, all the details on the machinery of MiCloud are dissected, compared with the other existing web platforms, MicrobiomeAnalyst [33], METAGENassist [34] and EzBioCloud [35], that intensely handle downstream data analysis rather than raw sequence data processing and microbiome profiling. In *Results*, we illustrate the use of MiCloud through the reanalysis of the United Kingdom (UK) twin data on the association between gut microbiome and body mass index (BMI) adjusting for age [2]. In *Discussion*, we discuss potential extensions and implementations of MiCloud.

## Materials and methods

MiCloud consists of three main components, named ***Data Processing***, ***Ecological Analysis*** and ***Taxonomical Analysis***, and many sub-components as in Fig 1. First, in ***Data Processing***, users can upload microbiome data in different formats through ***Data Input***, and then perform data filtering and quality controls through ***Quality Control***. Then, users can move to either ***Ecological Analysis*** or ***Taxonomical Analysis***. In ***Ecological Analysis***, users can calculate ecological measures (i.e., alpha diversity and beta diversity indices) through ***Diversity Calculation***, and then perform comparative/association analyses through ***Alpha Diversity*** and ***Beta Diversity***. In ***Taxonomical Analysis***, users can normalize taxonomic abundances in different ways through ***Data Transformation***, and then perform comparative/association analyses for microbial taxa in relative abundance through ***Comparison/Association***. MiCloud can handle all types of comparative/association analyses for a binary or continuous trait of interest, cross-sectional or longitudinal/family-based study design, and with or without covariate adjustment while the other existing web platforms cannot handle a continuous trait, longitudinal/family-based study design and covariate-adjusted analysis (Table 1).

There are many other statistical methods that can be considered for microbiome downstream data analysis, but the rationale for the selected statistical methods (Fig 1) is in their popularity, statistical validity and easy interpretation/presentation of the results as follows.

First, for the cross-sectional studies, statistical methods based on the independence assumption have been widely used. The Welch t-test and Wilcoxon rank-sum test [38] can be used for

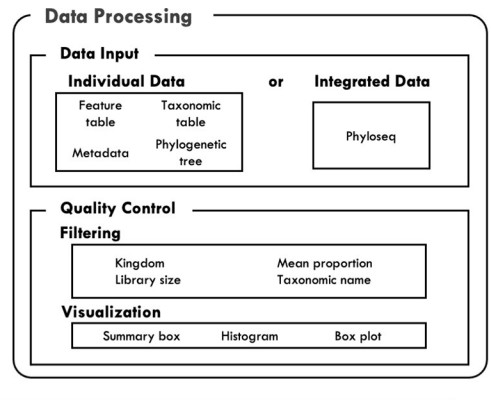

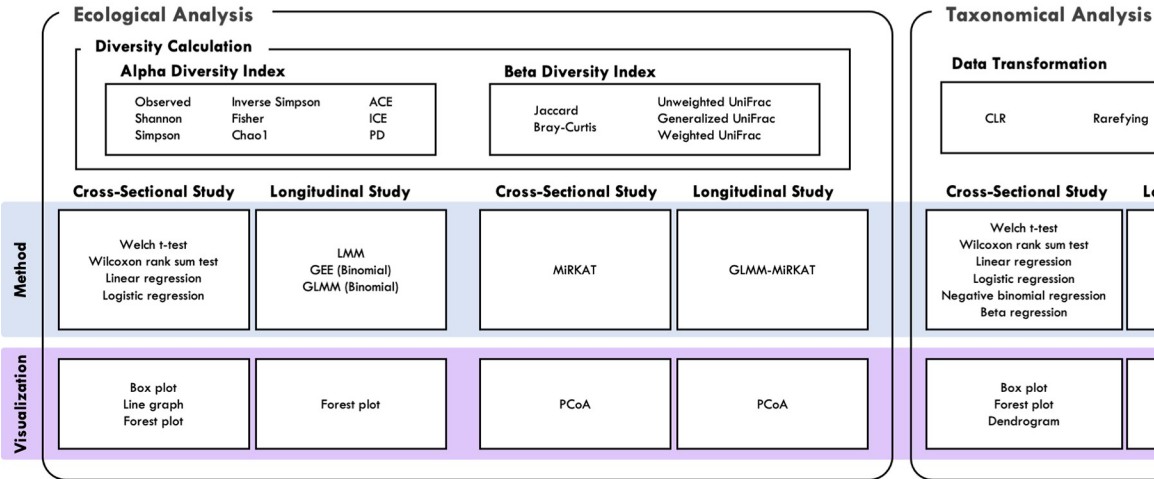

**Fig 1. Overview of MiCloud.** MiCloud consists of three main components, *Data Processing*, *Ecological Analysis* and *Taxonomical Analysis* and many sub-components.

**Table 1. The characteristics of MiCloud distinguished from the other existing web platforms, MicrobiomeAnalyst [33], METAGENassist [34] and EzBioCloud [35].**

|  | MiCloud | MicrobiomeAnalyst | METAGENassist | EzBioCloud |
|---|---|---|---|---|
| **Data Processing** | | | | |
| Data input | Individual & Integrated data | Individual data | Individual data | Raw sequence data |
| **Ecological Analysis** | | | | |
| **Alpha Diversity** | | | | |
| Longitudinal? | O | X | X | X |
| Continuous trait? | O | X | X | X |
| Covariate(s)? | O | X | X | X |
| **Beta Diversity** | | | | |
| Longitudinal? | O | X | X | X |
| Continuous trait? | O | X | X | X |
| Covariate? | O | X | X | X |
| **Taxonomical Analysis** | | | | |
| Longitudinal? | O | X | X | X |
| Continuous trait? | O | X | X | X |
| Covariate? | O | X | X | X |
| **Implementation Facility** | | | | |
| Local? | O | X | X | X |

non-covariate adjusted comparative analysis with a nice graphical presentation using box plots and summary statistics such as, mean, minimum, Q1, median, Q3 and maximum values. The linear regression, logistic regression, negative binomial regression, and beta regression models can be used for the continuous, binary, count and proportional response variables, respectively, with or without covariate adjustment, where the estimated regression coefficients, standard errors, confidence intervals, and P-values serve as a breadth of statistical inference facilities for the effect direction and size, variability and significance. The forest plot, line graph, and/or dendrogram can also efficiently summarize the results. Lastly, microbiome regression-based kernel association test (MiRKAT) [27, 28] has recently been highlighted for the beta-diversity analysis with or without covariate adjustment, where the principal coordinate analysis (PCoA) plot [39] can nicely summarize the results.

Second, in the longitudinal/family-based microbiome data, the repeated measurements from the same subject or the subjects from the same family tend to be correlated with each other because of the shared genetic components and environmental factors (e.g., diet, residence, etc). Hence, the statistical methods based on the independence assumption described above are not statistically valid, leading to inflated type I error rates, for longitudinal/family-based studies. Hence, we selected a series of statistical methods that are based on the random effects models (i.e., the linear mixed model (LMM) [40] and generalized linear mixed model (GLMM) [41]) or generalized estimating equations (GEE) [42] for both ecological and taxonomical analyses because of their well-known statistical validity (i.e., robust controls of type I error rate) for correlated data analysis. The results can also be presented using a breadth of statistical inference facilities, summary statistics and visualizations.

More details on each sub-component are addressed in following sections.

## Data processing: Data input

MiCloud requires four data components: feature table, taxonomic table, metadata, and phylogenetic tree. Users can upload them individually or in a single integrated format, called phyloseq [43]. The feature table is the count table where rows are OTUs or ASVs and columns are

subjects. Users can upload it in a tab-delimited text (.txt), comma-delimited text (.csv) or biological observation matrix (BIOM) format [44]. Especially, the BIOM format is the most widely used output format in many popular microbiome profiling pipelines, such as QIIME [16], MG-RAST [17], Mothur [18], MEGAN [19] and MetaPhlAn [20]; hence, users can directly upload it with no hassles. The taxonomic table should contain taxonomic names for microbial features (OTUs or ASVs) on seven taxonomic ranks, kingdom/domain, phylum, class, order, family, genus and species. Users can upload it in a tab-delimited text (.txt) or comma-delimited text (.csv) format. The metadata should contain variables for the subjects that are, for example, on host phenotypes, medical interventions, health/disease status, demographics, and so forth. Users can upload it in a tab-delimited text (.txt) or comma-delimited text (.csv) format. The phylogenetic tree represents evolutionary relationships across microbial features (OTUs or ASVs). Users can upload it in a Newick (.tre or.nwk) format. phyloseq is a well-organized microbiome data format that integrates all the four data components in a single R object, and it can be uploaded using a.rdata or.rds file. Once the data are uploaded, MiCloud verifies them before advancing to next steps. By default, MiCloud matches feature IDs and subject IDs across the four data components, and makes a rooted phylogenetic tree (if it is not rooted) through midpoint rooting [45].

Distinguished from the other existing web platforms, MiCloud can take the integrated data format, phyloseq (Table 1). EzBioCloud takes raw sequence data as inputs and performs microbiome profiling, while MiCloud, MicrobiomeAnalyst [33] and METAGENassist [34] do not (Table 1). Nephele [46], Qiita [47] and PUMAA [48] also take raw sequence data as inputs and perform comprehensive microbiome profiling for the 16S rRNA amplicon sequencing and/or shotgun metagenomics, yet they conduct only few exploratory downstream data analysis. For the raw sequence data processing and microbiome profiling, we recommend other popular and well-established bioinformatic pipelines, such as Nephele [46], QIIME2 (q2studio) [49], Qiita [47] and PUMAA [48] for web platforms, or QIIME [16], QIIME2 (q2cli) [49], MG-RAST [17], Mothur [18], MEGAN [19] and MetaPhlAn [20] for command line interfaces.

## Data processing: Quality control

MiCloud performs data filtering and quality controls in four criteria, kingdom, library size (i.e., total read count), mean proportion and taxonomic names as follows. Users can first type a kingdom of interest, which is, for example, Bacteria (default) for 16S data, Fungi for Internal Transcribed Spacer (ITS) [50] data or any other kingdom of interest for shotgun metagenomics. Then, users can remove subjects that have low library sizes (e.g., < 3,000 total read count (default)) and features (OTUs or ASVs) that have low mean proportions (e.g., < 0.002% (default)) using a slide bar. By default, MiCloud removes monotone and singleton features as they are likely to be sequencing errors and have almost no variation to be handled in downstream data analysis. Users can also remove erroneous taxonomic names in the taxonomic table that are completely or partially matched with the specified character strings, such as "uncultured", "incertae", "Incertae", "unidentified", "unclassified", "unknown", "metagenome", "gut metagenome", "mouse gut metagenome".

MiCloud visualizes microbiome data using summary boxes, histograms and box plots. The sample size and numbers of features (OTUs or ASVs), phyla, classes, orders, families, genera and species of the microbiome data are displayed in summary boxes. Library sizes across subjects and mean proportions across features are displayed in adjustable histograms and box plots. The graphs are updated in real-time to any changes in data filtering and quality controls. As such, users can interactively perform data filtering and quality controls. For additional

reference, MiCloud rarefies the count data to control varying library sizes [36]. The graphs and data after quality controls can be downloaded, where the graphs are especially in high resolution and appropriate size to be published.

## Ecological analysis: Diversity calculation

MiCloud performs ecological analyses in alpha diversity (a.k.a. within-sample diversity) and beta diversity (a.k.a. between-sample diversity). MiCloud calculates nine alpha diversity indices (i.e., Observed, Shannon [51], Simpson [52], Inverse Simpson [52], Fisher [53], Chao1 [54], abundance-based coverage estimator (ACE) [55], incidence-based coverage estimator (ICE) [56] and phylogenetic diversity (PD) [57]) and five beta diversity indices (i.e., Jaccard dissimilarity [58], Bray-Curtis dissimilarity [59], Unweighted UniFrac distance [60], Generalized UniFrac distance [61] and Weighted UniFrac distance [62]) (Fig 1). These indices are a proper mixture of richness and evenness, count and proportion with or without phylogenetic tree incorporation. MiCloud uses rarefied count data to calculate alpha diversity indices and count-based beta diversity indices (i.e., Jaccard dissimilarity [58] and Bray-Curtis dissimilarity [59]) because varying library sizes can heavily affect these indices [63]. For reference, the calculated diversity indices can be downloaded.

## Ecological analysis: Alpha diversity

MiCloud performs comparative/association analyses in alpha diversity. Users first need to click a tab for cross-sectional or longitudinal/family-based data analysis. More details on each are as follows.

### Cross-sectional (Fig 1)

Users first need to choose a primary variable that is a major trait of interest, such as host phenotypes, medical interventions and health/disease status, using a drop-down list. MiCloud automatically detects if it is binary or continuous. Then, MiCloud gives a chance to rename the categories (if it is binary) or the variable name (if it is continuous) to be appropriately displayed in later graphs. Then, users can choose covariates, such as age and gender, or not. Then, MiCloud lists statistical methods as follows. For a binary trait with no covariates, the Welch t-test and Wilcoxon rank-sum test [38] are listed. For a binary trait with covariates, the linear regression (with each alpha diversity index as a response, and the primary variable as a predictor) and the logistic regression (with the primary variable as a response, and each alpha diversity index as a predictor) are listed. For a continuous trait with or without covariates, the linear regression (with each alpha diversity index as a response, and the primary variable as a predictor) is listed. Lastly, users can address the multiplicity issue or not. For the multiple testing adjustment, the Benjamini-Hochberg (BH) procedures [64] can be employed to control false discovery rate (FDR).

### Longitudinal (Fig 1)

All the widgets for the cross-sectional data analysis (i.e., primary variable, rename categories/variable, covariate(s), method and multiple testing adjustment) are retained for the longitudinal/family-based data analysis, yet there are some additional widgets for the longitudinal/family-based data analysis as follows. First, users need to choose a cluster variable that contains, for example, subject IDs for repeated measurements or family IDs for family-based studies. Second, MiCloud lists statistical methods as follows. For a binary trait with or without covariates, LMM [40] (with each alpha diversity index as a response, and the primary variable as a

predictor), GEE (Binomial) [42] and GLMM (Binomial) [41] (with the primary variable as a response, and each alpha diversity index as a predictor) are listed. For a continuous trait with or without covariates, LMM (with each alpha diversity index as a response, and the primary variable as a predictor) is listed.

MiCloud visualizes the results using box plots, line graphs or forest plots, calculates summary statistics, and organizes them in output tables. The graphs (by clicking the right mouse button on the plot then through "Save Image as") and output tables can be downloaded, and the graphs are in high resolution and appropriate size to be published.

## Ecological analysis: Beta diversity

MiCloud performs comparative/association analyses in beta diversity. As in alpha diversity analysis, users first need to click a tab for cross-sectional or longitudinal/family-based data analysis. More details on each are as follows.

### Cross-sectional (Fig 1)

All the widgets for the cross-sectional data analysis in alpha diversity (i.e., primary variable, rename categories/variable, covariate(s) and method) are retained for the cross-sectional data analysis in beta diversity. Yet, in methodology, MiRKAT [27, 28] is listed.

### Longitudinal (Fig 1)

The widgets in the longitudinal/family-based data analysis in alpha diversity (i.e., primary variable, rename categories/variable, cluster variable, covariate(s) and method) are retained in the longitudinal/family-based data analysis in beta diversity. Yet in methodology, generalized linear mixed model—microbiome regression-based kernel association test (GLMM-MiRKAT) [28, 65] is listed.

The results are visualized using PCoA plots [39]. Again, the graphs (by clicking the right mouse button on the plot then through "Save Image as") and output tables can be downloaded, and the graphs are in high resolution and appropriate size to be published.

## Taxonomical analysis: Data transformation

MiCloud considers four commonly used taxonomic abundance forms of count, rarefied count [36], proportion and CLR [37]. For the CLR transformation, MiCloud replaces zeros with non-zero values using the Bayesian multiplicative replacement [66]. For reference, users can download all the four data forms.

## Taxonomical analysis: Comparison/association

MiCloud performs comparative/association analysis for microbial taxa in relative abundance on different taxonomic levels (i.e., phylum, class, order, family, genus and species). As in ecological analysis, users first need to click a tab for cross-sectional or longitudinal/family-based data analysis. More details on each are as follows.

### Cross-sectional (Fig 1)

The widgets for the cross-sectional data analysis in alpha diversity (i.e., primary variable, rename categories/variable, covariate(s) and method) are retained in the cross-sectional data analysis for microbial taxa in relative abundance, yet there are some additional widgets as follows. First, users need to choose a data form among CLR (default), count and proportion. Second, MiCloud lists statistical methods that are suited to the chosen data form as follows.

i. *CLR*. For a binary trait without covariates, the Welch t-test, Wilcoxon rank-sum test [38], linear regression (with each taxon as a response, and the primary variable as a predictor) and logistic regression (with the primary variable as a response, and each taxon as a predictor) are listed. For a binary trait with covariates, the linear regression (with each taxon as a response, and the primary variable as a predictor) and the logistic regression (with the primary variable as a response, and each taxon as a predictor) are listed. For a continuous trait with or without covariates, the linear regression (with the primary variable as a response, and each taxon as a predictor) is listed.

ii. *Count*. For a binary trait without covariates, the Welch t-test, Wilcoxon rank-sum test [38] and logistic regression (with the primary variable as a response, and each taxon as a predictor) using rarefied count data, and the negative binomial regression (with each taxon as a response, and the primary variable as a predictor) using original count data with the library size (total read count) as an offset variable are listed. For a binary trait with covariates, the logistic regression (with the primary variable as a response, and each taxon as a predictor) using rarefied count data, and the negative binomial regression (with each taxon as a response, and the primary variable as a predictor) using original count data with the library size (total read count) as an offset variable are listed. For a continuous trait with or without covariates, the negative binomial regression (with each taxon as a response, and the primary variable as a predictor) using original count data with the library size (total read count) as an offset variable is listed.

iii. *Proportion*. For a binary trait without covariates, the Welch t-test, Wilcoxon rank-sum test [38], logistic regression (with the primary variable as a response, and each taxon as a predictor) and beta regression (with each taxon as a response, and the primary variable as a predictor) are listed. For a binary trait with covariates, the logistic regression (with the primary variable as a response, and each taxon as a predictor), and beta regression (with each taxon as a response, and the primary variable as a predictor) are listed. For a continuous trait with or without covariates, the beta regression (with each taxon as a response, and the primary variable as a predictor) is listed.

## Longitudinal (Fig 1)

The widgets in the cross-sectional data analysis for microbial taxa (i.e., primary variable, rename categories/variable, covariate(s) and method) are retained in the longitudinal/family-based data analysis for microbial taxa, yet there are some additional widgets as follows. First, users need to choose a cluster variable that contains, for example, subject IDs for repeated measures or family IDs for family-based studies. Second, MiCloud lists different statistical methods as follows.

i. *CLR*. For a binary trait with or without covariates, LMM [40] (with each taxon as a response, and the primary variable as a predictor), GLMM (Binomial) [41] (with the primary variable as a response, and each taxon as a predictor) and GEE (Binomial) [42] (with the primary variable as a response, and each taxon as a predictor) are listed. For a continuous trait with or without covariates, LMM [40] (with each taxon as a response, and the primary variable as a predictor) is listed.

ii. *Count*. For a binary trait with or without covariates, GLMM (Binomial) [41] (with the primary variable as a response, and each taxon as a predictor) and GEE (Binomial) [42] (with the primary variable as a response, and each taxon as a predictor) using rarefied count data,

and GLMM (Negative Binomial) [41] (with each taxon as a response, and the primary variable as a predictor) using original count data with the library size (total read count) as an offset variable are listed. For a continuous trait with or without covariates, GLMM (Negative Binomial) [41] (with each taxon as a response, and the primary variable as a predictor) using original count data with the library size (total read count) as an offset variable is listed.

iii. *Proportion.* For a binary trait with or without covariates, GLMM (Binomial) [41] (with the primary variable as a response, and each taxon as a predictor), GEE (Binomial) [42] (with the primary variable as a response, and each taxon as a predictor) and GLMM (Beta) [41] (with each taxon as a response, and the primary variable as a predictor) are listed. For a continuous trait with or without covariates, the GLMM (Beta) [41] (with each taxon as a response, and the primary variable as a predictor) is listed.

We note that the use of the rarefied count data or the original count data with the library size (total read count) as an offset variable is to account for varying library sizes (total read counts) due to uneven sequencing depths across subjects when the count data form is employed. Users can perform taxonomical analyses from phylum to genus (e.g., for 16S data) or from phylum to species (e.g., for shotgun metagenomics). For the multiple testing adjustment, MiCloud applies the BH procedures [64] to control FDR per taxonomic level. MiCloud visualizes the results using box plots, forest plots, and dendrograms. Especially, the dendrogram presents the hierarchical discovery status using colors (red: positive association, blue: negative association, gray: non-significance). Again, the graphs and output tables can be downloaded, and the graphs are in high resolution and appropriate size to be published.

## Web server and local implementation

We wrote MiCloud in R language using the R package, called Shiny (https://shiny.rstudio.com/), and deployed the web application using ShinyProxy (https://www.shinyproxy.io/). Currently, the web server has the specification of Intel Core i7 processor (8 cores, 2.90–4.80 GHz) and 36 GB DDR4 memory, and supports up to ten concurrent users. We are committed to monitoring the usage, performance and availability of the web server periodically to maintain it stable. However, in case that the web server is too busy, we created the GitHub repository that enables local implementation on the user's computer, while the other existing web platforms can be implemented only on the web server (Table 1). The URLs are http://micloud.kr (web application) and https://github.com/wg99526/micloudgit (GitHub).

## Results

Here, we illustrate the use of MiCloud through the reanalysis of the UK twin study data [2] on the association between gut microbiome and BMI adjusting for age. This example illustration is for a continuous trait of interest (BMI), family-based study design (twin study) and covariate-adjusted (age-adjusted) analysis, which cannot be handled by the other existing web platforms.

Goodrich et al. (2014) collected fecal samples from the UK twin population, and then profiled their microbiomes targeting the V4 region of the 16S rRNA gene [2]. The raw sequence data are publicly available in the European Bioinformatics Institute (EMBL-EBI) database (access number: ERP006339 and ERP006342) [2]. We processed the raw sequence data using QIIME [16], and acquired the feature table and taxonomic table using open-reference OTU picking with 97% sequence similarity, and the phylogenetic tree using FastTree [67]. The original microbiome data we used consist of 7349 OTUs, 17 phyla, 31 classes, 60 orders, 105

families, 232 genera and 173 species for 370 monozygotic twins. We stored them as example 16S data in the phyloseq format [43] on MiCloud. The rest of the data processing and analytic procedures are as follows.

We uploaded the data in the phyloseq format [43], and then performed data filtering and quality controls using default settings. Then, 1622 OTUs, 10 phyla, 18 classes, 25 orders, 41 families, 77 genera and 55 species for 370 monozygotic twins were retained. The library sizes across subjects and the mean proportions across OTUs are visualized in histograms and box plots (S1 and S2 Figs). There, we can observe varying library sizes (S1 Fig) and highly skewed mean proportions (S2 Fig).

We performed family-based data analyses for ecological measures (i.e., alpha and beta diversity indices) and microbial taxa from phyla to genera, while setting BMI as the primary variable, family ID as the cluster variable and age as a covariate. We fitted LMM [40] for alpha diversity analysis (Fig 2) and GLMM-MiRKAT [28, 65] for beta diversity analysis (Fig 3). Then, we found negative associations between BMI and seven alpha diversity indices (Observed, Shannon [51], Fisher [53], Chao1 [54], ACE [55], ICE [56] and PD [57]) at the significance level of 5%, yet the results for the Simpson [52] and Inverse Simpson [52] indices are not statistically significant (Fig 2). We also found significant associations between BMI and four beta diversity indices (i.e., Jaccard dissimilarity [58], Bray-Curtis dissimilarity [59], Unweighted UniFrac distance [60] and Generalized UniFrac distance [61]) at the significance level of 5%, yet the result for the Weighted UniFrac distance [62] is not statistically significant (Fig 3). aGLMM-MiRKAT, that is the significance test that combines all the results from the five beta diversity indices, shows a significant association between BMI and beta diversity (Fig 3). Lastly, for taxonomical analysis, we fitted LMM using CLR transformed data. Then, we found 1) positive associations between BMI and two phyla (Firmicutes and Actinobacteria), three classes (Bacilli, Clostridia and Actinobacteria), two orders (Lactobacillates and

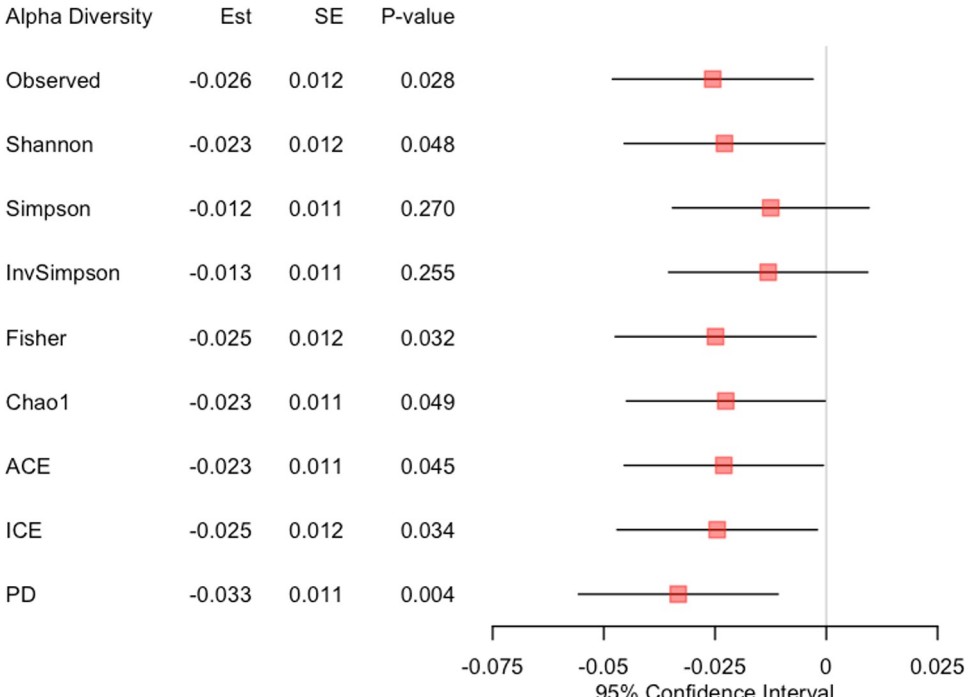

**Fig 2. The results for alpha diversity analysis.** Est represents the estimated coefficient, and SE represents the standard error.

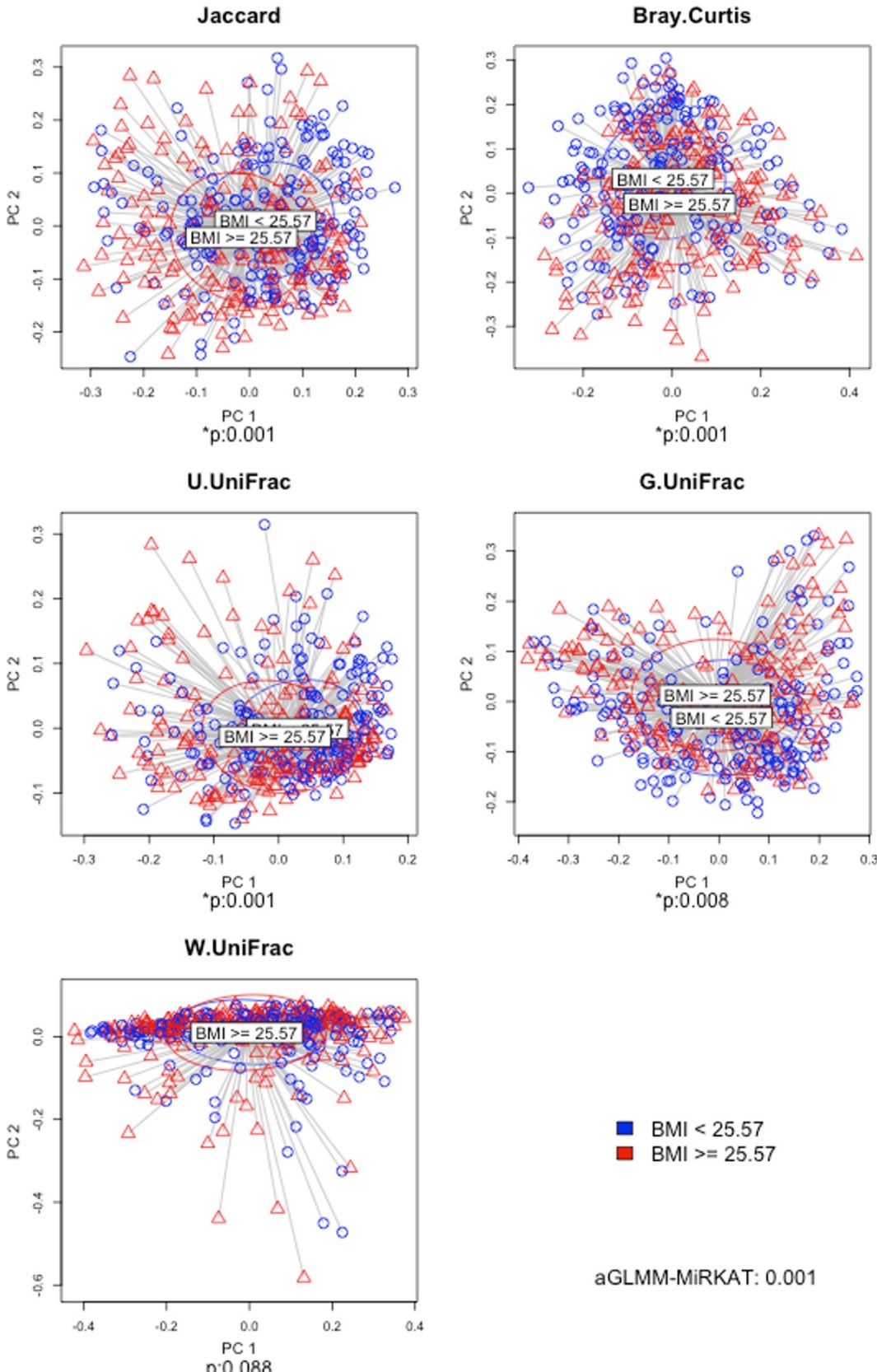

**Fig 3. The results for beta diversity analysis.** The two-dimensional PCoA plots visualize each beta diversity index stratified by two BMI categories (i.e., BMI < 25.57 and BMI ≥ 25.57, where 25.57 is the median BMI). *p represents the P-values estimated by GLMM-MiRKAT [28, 65]. Jacaard: Jaccard dissimilarity [58]. BC: Bray-Curtis dissimilarity [59]. U.UniFrac: Unweighted UniFrac distance [60]. G.UniFrac: Generalized UniFrac distance [61]. W.UniFrac: Weighted UniFrac disance [62].

Actinomycetales), three families (Streptococcaeae, Lactobacillaceae and Actinomycetaceae) and three genera (Streptococcus, Lactobacillus and Acidaminococcus), and 2) negative association between BMI and one phylum (Tenericules), two classes (Mollicutes and RF3), two orders (ML615J-28 and RF 39) and two families (Christensenellaceae and S24-7) at the significance level of 5% after addressing the multiplicity issue using the BH procedures [64] (Figs 4 and 5).

## Discussion

In this paper, we introduced MiCloud for comprehensive microbiome data analysis on user-friendly web environments. MiCloud enables comparative/association analysis for a binary or continuous trait of interest, cross-sectional or longitudinal/family-based study design, and with or without covariate adjustment while other existing web platforms cannot handle a continuous trait, longitudinal/family-based study design and covariate-adjusted analysis. Especially, in the longitudinal/family-based microbiome data, the repeated measurements from the same subject or the subjects from the same family tend to be correlated with each other due to the shared genetic components and environmental factors. Hence, the statistical methods based on the independence assumption, used in other existing web platforms or used for cross-sectional studies in MiCloud, are not statistically valid, leading to inflated type I error rates. However, MiCloud employs, in addition, a series of statistical methods that are based on the random effects models [40] or GEE [42] (Table 1) for both ecological and taxonomical analyses; as such, users can easily handle correlated data from longitudinal/family-based microbiome studies on our user-friendly web environments. We demonstrated the use of MiCloud through the reanalysis of the UK twin study data [2] for a continuous trait of interest (i.e., BMI), family-based study design and covariate-adjusted (i.e., age-adjusted) analysis that cannot be handled by other existing web platforms.

We used R Shiny to develop MiCloud. Many of the current statistical methods and visualization approaches are written in R language, and they are freely available through R libraries;

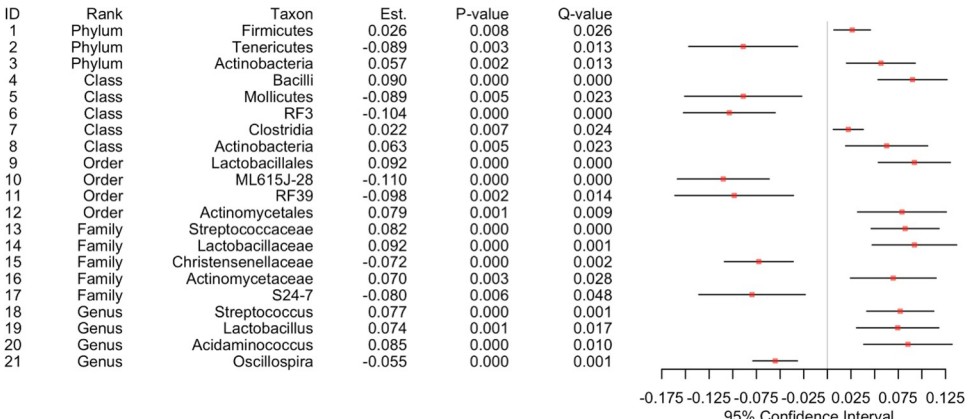

**Fig 4. The results for taxonomical analysis in forest plot.** Est represents the estimated coefficient, and Q-value represents the FDR-adjusted P-value.

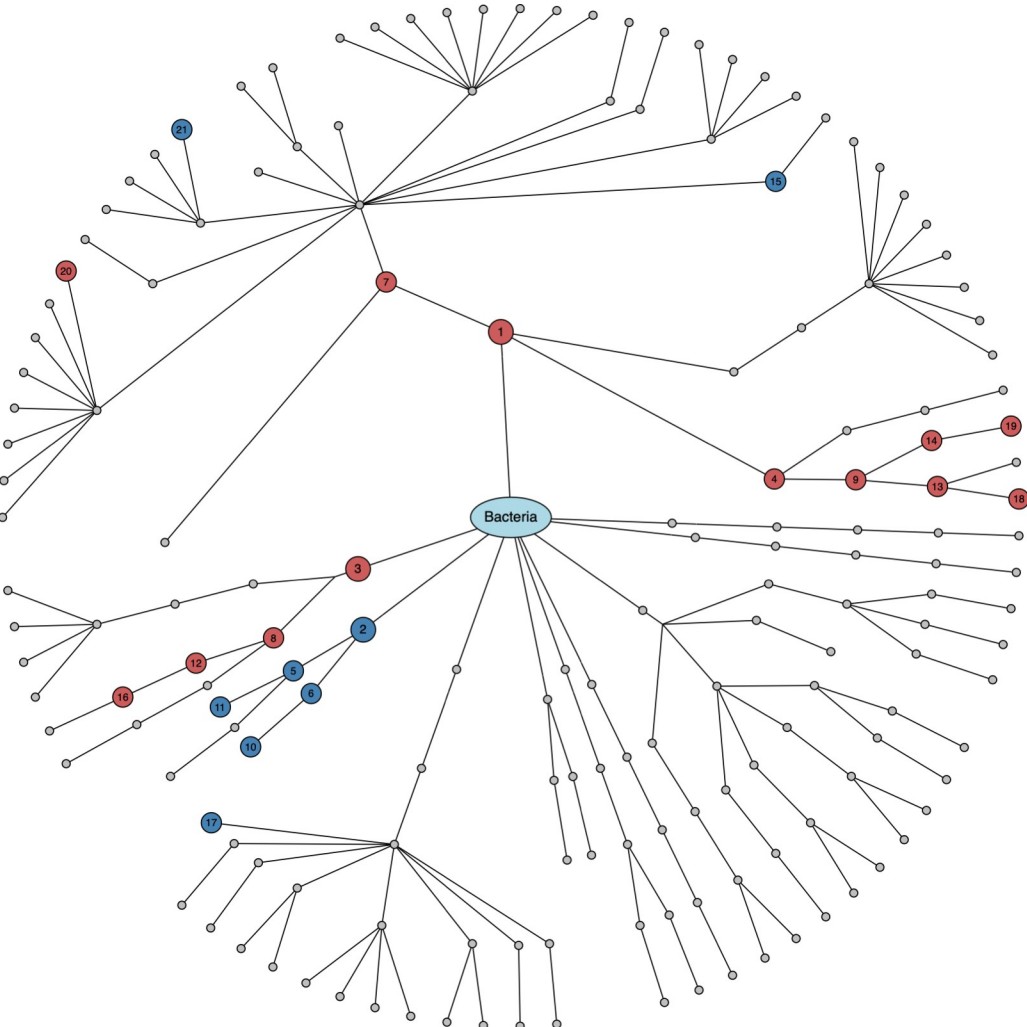

**Fig 5. The results for taxonomical analysis in dendrogram.** The numbers in circles are the IDs in the forest plot (Fig 4). Red: positive association. Blue: negative association. Gray: non-significance.

hence, we could easily transfer them to MiCloud. Galaxy [68] is another popular platform to develop web applications in computational biology, but many of the current Galaxy applications are written in different programming languages, and focus more on raw sequence data processing and genome/microbiome profiling, rather than downstream data analysis. It is beyond the scope of this research to compare R Shiny to Galaxy, but we would say that R Shiny can be better for downstream data analysis, while Galaxy can be better for upstream data processing.

We also elaborated in many other facilities, such as data inputs (individual and integrated data forms), quality controls (with respect to kingdom, library size, mean proportion and taxonomic name), data transformations (various alpha and beta diversity indices, and taxonomic abundance forms of count, rarefied count [36], proportion and CLR [37]), statistical methods (various methods for different study designs, data forms and analytic schemes), visualizations (various plots for data summary and ecological/taxonomical analyses) and implementations (on the web server or user's computer). Hence, users in various disciplines, even non-professional programmers like clinicians and biologists, can flexibly perform microbiome data

analysis. All the normalized data, output tables and graphs generated by MiCloud are down-loadable and/or publishable; hence, it is straightforward to present or reanalyze the results.

However, we note here that in microbiome studies, researchers have performed more types of data analysis with different aims and data forms, such as prediction analysis, gene-level/functional analysis, multivariate analysis, survival analysis, time-series analysis, and so forth. MiCloud extends web-based microbiome data analytics to covariate-adjusted analysis and longitudinal/family-based data analysis, yet MiCloud does not handle upstream data processing (raw sequence data processing and microbiome profiling) and all possible types of downstream data analysis. Further extensions of MiCloud are therefore needed for more comprehensive microbiome data analysis.

## Supporting information

**S1 Fig.** The histogram (A) and box plot (B) for library sizes across subjects after quality controls.
(TIF)

**S2 Fig.** The histogram (A) and box plot (B) for mean proportions across OTUs after quality controls.
(TIF)

## Author Contributions

**Conceptualization:** Hyunwook Koh.

**Data curation:** Hyunwook Koh.

**Formal analysis:** Won Gu, Jeongsup Moon, Crispen Chisina, Hyunwook Koh.

**Funding acquisition:** Taesung Park, Hyunwook Koh.

**Investigation:** Jeongsup Moon, Byungkon Kang, Taesung Park, Hyunwook Koh.

**Methodology:** Won Gu, Taesung Park, Hyunwook Koh.

**Project administration:** Taesung Park.

**Resources:** Taesung Park, Hyunwook Koh.

**Software:** Won Gu, Jeongsup Moon, Crispen Chisina, Byungkon Kang, Hyunwook Koh.

**Supervision:** Byungkon Kang, Taesung Park, Hyunwook Koh.

**Validation:** Taesung Park, Hyunwook Koh.

**Visualization:** Won Gu, Jeongsup Moon, Crispen Chisina, Hyunwook Koh.

**Writing – original draft:** Won Gu, Jeongsup Moon, Hyunwook Koh.

**Writing – review & editing:** Won Gu, Jeongsup Moon, Crispen Chisina, Byungkon Kang, Taesung Park, Hyunwook Koh.

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
