## [Decision Letter · Decision Letter 0]

18 May 2022

PONE-D-22-09539MiCloud: A unified web platform for comprehensive microbiome data analysisPLOS ONE

Dear Dr. Koh,

Thank you for submitting your manuscript to PLOS ONE. After careful consideration, we feel that it has merit but does not fully meet PLOS ONE’s publication criteria as it currently stands. Therefore, we invite you to submit a revised version of the manuscript that addresses the points raised by the reviewer and by myself.

My first comment concerns the validation of the findings provided by MiCloud. You have used a dataset of a twin study to show what your pipeline is able to do and you presented the obtained data. But, there is no validation of this data for example by comparing these findings with those obtained on the same dataset by using another analysis tool.

My second comments concern the discussion, which is very short. Knowing that MiCloud seems to perform a restricted number (but very useful) analyses that cannot be handle by other existing web platform, I ask me for example why you have chosen to develop a new platform and not to improve one of these existing platform? More globally, I ask you to discuss more in deep, the advantages and the limits of your tool.

We look forward to receiving your revised manuscript.

Kind regards,

Jean-François Humbert

Academic Editor

PLOS ONE

Journal Requirements:

Reviewers' comments:

Reviewer's Responses to Questions

**Comments to the Author**

1. Is the manuscript technically sound, and do the data support the conclusions?

Reviewer #1: Yes

2. Has the statistical analysis been performed appropriately and rigorously? 

Reviewer #1: Yes

3. Have the authors made all data underlying the findings in their manuscript fully available?

Reviewer #1: No

4. Is the manuscript presented in an intelligible fashion and written in standard English?

Reviewer #1: Yes

5. Review Comments to the Author

Reviewer #1: The paper describes MiCloud, an online computer application that aims to expand beyond the functionality of existing web platforms that focus on categorical traits, cross-sectional study design, and no covariate adjustment. The additional functionality provided by the web-service is useful for the community to fill in these gaps. The system enables users to perform processing and statistical analyses on microbiome data, especially providing ways to analyze longitudinal data, continuous and data with covariates. The authors implemented the statistical analyses tools using R Shiny, making it easy for users to run on a server or local compute.

Overall, the paper is sound and seems to address a need for more advanced statistical analysis (at least for cross-sectional and longitudinal). It mentions in discussion several other areas (e.g. survival, etc.) that it can’t currently handle but would like to address in future.

Comments

(1) The GitHub Repo https://github.com/wg99526/micloud doesn’t appear to be available but the web server hosting MiCloud was available http://223.194.200.160:3838/ .

The closest I found of the Repo was https://github.com/wg99526/MiCloudGit The final repo should have information/instruction in their README on how to run this in local computer.

This sentence makese me wonder how supported the system is and whether I can rely on it: "In case that the web server is too busy, we created the GitHub repository that enables local implementation on the user’s computer, while the other existing web platforms can be implemented only on the web server (Table 1)."

(2) The authors describe the data input and data processing methods in good detail. They then proceed to describe the statistical methods available for ecological analyses and finally discuss the available tools for taxonomical analysis. The last sentence before Taxonomical Analysis, says “The results are visualized using principal coordinate analysis (PCoA) plots (Torgerson, 1952). Again, the graphs and output tables can be downloaded, and the graphs are in high resolution and appropriate size to be published.” This is not entirely accurate because in the tool, while a table of beta diversity results could be exported, the images were not available for download directly from the tool.

(3) Authors listed and referenced the statistical methods used for analysis as well as appropriate data transformation methods applicable to specific tools. It was nice to see references in the actual R Shiny as well.

(4) In this sentence: “All the normalized data, output tables and graphs generated by MiCloud are adjustable, downloadable and/or publishable; hence, it is straightforward to present or reanalyze the results.” – the use of “adjustable” doesn’t seem always the case as only the quality control plots are interactive with the use of plotly.

(5) While the choice of methods for the cross-sectional and longitudinal methods appears to be sound, the rationale for the choice is not elaborated in the methods and could benefit from some additional references and description of why the methodology is most suitable. Moreover, the discussion section could include a comparison of the selected methods and the benefits/disadvantages to other alternatives for these use cases such as cross-sectional versus longitudinal modeling.

(6) The manuscript introduction could benefit from a more thorough referencing of available web-based services relating to microbiome analysis as a few are excluded. For example, in the last sentence of “Data Processing: Data Input” consider citing Nephele as a cloud plaform for raw sequence data processing and microbiome profiling.

Weber N., et al. (2018) Nephele: a cloud platform for simplified, standardized and reproducible microbiome data analysis. Bioinformatics, 34(8): 1411–1413. https://doi.org/10.1093/bioinformatics/btx617

See also: Gonzalez et al. [Qiita], Mitchell et al. [PUMAA]).

In fact, it would be helpful for MiCloud’s users if they add “external resource” information on their site for those who wonder how they prepare the input files for MiCloud, e.g. "For the raw sequence data processing and microbiome profiling, we recommend other popular and well-established bioinformatic pipelines, such as online platform Nephele, Galaxy, QIIME2, etc..."

6. PLOS authors have the option to publish the peer review history of their article (what does this mean?). If published, this will include your full peer review and any attached files.

Reviewer #1: No

---

## [Author Response · Author response to Decision Letter 0]

27 Jun 2022

Response Letter

Reviewer’s Comments

The paper describes MiCloud, an online computer application that aims to expand beyond the functionality of existing web platforms that focus on categorical traits, cross-sectional study design, and no covariate adjustment. The additional functionality provided by the web-service is useful for the community to fill in these gaps. The system enables users to perform processing and statistical analyses on microbiome data, especially providing ways to analyze longitudinal data, continuous and data with covariates. The authors implemented the statistical analyses tools using R Shiny, making it easy for users to run on a server or local compute. Overall, the paper is sound and seems to address a need for more advanced statistical analysis (at least for cross-sectional and longitudinal). It mentions in discussion several other areas (e.g. survival, etc.) that it can’t currently handle but would like to address in future.

(Response) Thank you very much for your careful observations and insightful comments. They have made much improvement to the manuscript and web platform. We have responded your comments below, and updated the app and manuscript, accordingly. 

Comments:

1. The GitHub Repo https://github.com/wg99526/micloud doesn’t appear to be available but the web server hosting MiCloud was available http://223.194.200.160:3838/. The closest I found of the Repo was https://github.com/wg99526/micloudgit. The final repo should have information/instruction in their README on how to run this in local computer. This sentence makes me wonder how supported the system is and whether I can rely on it: "In case that the web server is too busy, we created the GitHub repository that enables local implementation on the user’s computer, whilethe other existing web platforms can be implemented only on the web server (Table 1)."

(Response) We also found that the URL for the local implementation (GitHub) was wrong, and the information/instruction was confusing. We apologize for the inconvenience caused. We have first corrected the URL for the GitHub repository to https://github.com/wg99526/micloudgit in the revised manuscript, and updated the online manual (readme) including 1) description, 2) URLs, 3) references, 4) prerequites, 5) launch app, 6) troubleshooting tips, and 7) external resources. Users should be able to run it by simply following the step-by-step instructions. We also described current specification and capacity of the web server in the revised manuscript as follows. “Currently, the web server has the specification of Intel Core i7 processor (8 cores, 2.90-4.80 GHz) and 36 GB DDR4 memory, and supports up to ten concurrent users. We are committed to monitoring the usage, performance and availability of the web server periodically to maintain it stable.” In addition, we bought a domain name, http://micloud.kr, for the web application URL.

2. The authors describe the data input and data processing methods in good detail. They then proceed to describe the statistical methods available for ecological analyses and finally discuss the available tools for taxonomical analysis. The last sentence before Taxonomical Analysis, says “The results are visualized using principal coordinate analysis (PCoA) plots (Torgerson, 1952). Again, the graphs and output tables can be downloaded, and the graphs are in high resolution and appropriate size to be published.” This is not entirely accurate because in the tool, while a table of beta diversity results could be exported, the images were not available for download directly from the tool.

(Response) The PCoA plot can be downloaded by clicking the right mouse button on the plot and then through “Save Image As”. We revised the manuscript as follows.

“The results are visualized using PCoA plots (Torgerson, 1952). Again, the graphs (by clicking the right mouse button on the plot then through “Save Image as”) and output tables can be downloaded, and the graphs are in high resolution and appropriate size to be published.”

3. Authors listed and referenced the statistical methods used for analysis as well as appropriate data transformation methods applicable to specific tools. It was nice to see references in the actual R Shiny as well.

(Response) Thank you very much for your positive comments.

4. In this sentence: “All the normalized data, output tables and graphs generated by MiCloud are adjustable, downloadable and/or publishable; hence, it is straightforward to present or reanalyze the results.” - the use of “adjustable” doesn’t seem always the case as only the quality control plots are interactive with the use of plotly.

(Response) We deleted “adjustable” in the manuscript as follows.

“All the normalized data, output tables and graphs generated by MiCloud are downloadable and/or publishable; hence, it is straightforward to present or reanalyze the results.”

5. While the choice of methods for the cross-sectional and longitudinal methods appears to be sound, the rationale for the choice is not elaborated in the methods and could benefit from some additional references and description of why the methodology is most suitable. Moreover, the discussion section could include a comparison of the selected methods and the benefits/disadvantages to other alternatives for these use cases such as cross-sectional versus longitudinal modeling. 

(Response) We do not propose any new statistical methods in this paper. MiCloud is a web platform that makes user-friendly implementation of existing methods that have been only available through command line interfaces. Thus, we referenced the original articles for more details. However, we described the rationale for the selected statistical methods in the Materials and Methods as follows.

“There are many other statistical methods that can be considered for microbiome downstream data analysis, but the rationale for the selected statistical methods (Fig 1) is in their popularity, statistical validity and easy interpretation/presentation of the results as follows. 

First, for the cross-sectional studies, statistical methods based on the independence assumption have been widely used. The Welch t-test and Wilcoxon rank-sum test (Mann and Whitney, 1947) can be used for non-covariate adjusted comparative analysis with a nice graphical presentation using box plots and summary statistics such as, mean, minimum, Q1, median, Q3 and maximum values. The linear regression, logistic regression, negative binomial regression, and beta regression models can be used for the continuous, binary, count and proportional response variables, respectively, with or without covariate adjustment, where the estimated regression coefficients, standard errors, confidence intervals, and P-values serve as a breadth of statistical inference facilities for the effect direction and size, variability and significance. The forest plot, line graph, and/or dendrogram can also efficiently summarize the results. Lastly, microbiome regression-based kernel association test (MiRKAT) (Zhao et al., 2015, Wilson et al., 2021) has recently been highlighted for the beta-diversity analysis with or without covariate adjustment, where the principal coordinate analysis (PCoA) plot (Torgerson, 1952) can nicely summarize the results.

Second, in the longitudinal/family-based microbiome data, the repeated measurements from the same subject or the subjects from the same family tend to be correlated with each other because of the shared genetic components and environmental factors (e.g., diet, residence, etc). Hence, the statistical methods based on the independence assumption described above are not statistically valid, leading to inflated type I error rates, for longitudinal/family-based studies. Hence, we selected a series of statistical methods that are based on the random effects models (i.e., the linear mixed model (LMM) (Laird and Ware, 1982) and generalized linear mixed model (GLMM) (Breslow and Clayton, 1993)) or generalized estimating equations (GEE) (Liang and Zeger, 1986) for both ecological and taxonomical analyses because of their well-known statistical validity (i.e., robust controls of type I error rate) for correlated data analysis. The results can also be presented using a breadth of statistical inference facilities, summary statistics and visualizations. 

More details on each sub-component are addressed in following sections.”

We also included some description on cross-sectional vs. longitudinal methods in the Discussion as follows. 

“Especially, in the longitudinal/family-based microbiome data, the repeated measurements from the same subject or the subjects from the same family tend to be correlated with each other due to the shared genetic components and environmental factors. Hence, the statistical methods based on the independence assumption, used in other existing web platforms or used for cross-sectional studies in MiCloud, are not statistically valid, leading to inflated type I error rates. However, MiCloud employs, in addition, a series of statistical methods that are based on the random effects models (Laird and Ware, 1982) or GEE (Liang and Zeger, 1986) (Table 1) for both ecological and taxonomical analyses; as such, users can easily handle correlated data from longitudinal/family-based microbiome studies on our user-friendly web environments.”

6. The manuscript introduction could benefit from a more thorough referencing of available web-based services relating to microbiome analysis as a few are excluded. For example, in the last sentence of “Data Processing: Data Input” consider citing Nephele as a cloud plaform for raw sequence data processing and microbiome profiling. Weber N., et al. (2018) Nephele: a cloud platform for simplified, standardized and reproducible microbiome data analysis. Bioinformatics, 34(8): 1411–1413. 

https://doi.org/10.1093/bioinformatics/btx617 See also: Gonzalez et al. [Qiita], Mitchell et al. [PUMAA]).

In fact, it would be helpful for MiCloud’s users if they add “external resource” information on their site for those who wonder how they prepare the input files for MiCloud, e.g. "For the raw sequence data processing and microbiome profiling, we recommend other popular and well-established bioinformatic pipelines, such as online platform Nephele, Galaxy, QIIME2, etc..."

(Response) MiCloud handles downstream data analysis. Thus, we compared MiCloud with MicrobiomeAnalyst (Dhariwal et al., 2017), METAGENassist (Arndt et al., 2012) and EzBioCloud (Yoon et al., 2017) that intensely handle downstream data analysis rather than raw sequence data processing and microbiome profiling. We also surveyed Nephele, Qiita and PUMAA, and found that they are mostly for raw sequence data processing and microbiome profiling (they conduct only some exploratory data analysis in different contexts). Thus, it was difficult to directly compare them with MiCloud. 

We revised the Introduction section to better clarify it as follows.

“However, existing web platforms for downstream microbiome data analysis, including MicrobiomeAnalyst (Dhariwal et al., 2017), METAGENassist (Arndt et al., 2012) and EzBioCloud (Yoon et al., 2017), can handle only a categorical trait of interest (e.g., diseased vs. healthy, treatment vs. placebo), cross-sectional study, and the analysis with no covariate adjustment.” 

“In Materials and Methods, all the details on the machinery of MiCloud are dissected, compared with the other existing web platforms, MicrobiomeAnalyst (Dhariwal et al., 2017), METAGENassist (Arndt et al., 2012) and EzBioCloud (Yoon et al., 2017), that intensely handle downstream data analysis rather than raw sequence data processing and microbiome profiling.”

We also revised the Data Processing: Data Input section as follows.

“Nephele (Weber et al., 2018), Qiita (Gonzalez et al., 2018) and PUMAA (Mitchell et al., 2020) also take raw sequence data as inputs and perform comprehensive microbiome profiling for the 16S rRNA amplicon sequencing and/or shotgun metagenomics, yet they conduct only some exploratory downstream data analysis. For the raw sequence data processing and microbiome profiling, we recommend other popular and well-established bioinformatic pipelines, such as Nephele (Weber et al., 2018), QIIME2 (q2studio) (Bolyen et al., 2019), Qiita (Gonzalez et al., 2018) and PUMAA (Mitchell et al., 2020) for web platforms, or QIIME (Caporaso et al., 2010), QIIME2 (q2cli) (Bolyen et al., 2019), MG-RAST (Meyer et al., 2008), Mothur (Schloss et al., 2009), MEGAN (Huson et al., 2007) and MetaPhlAn (Segata et al., 2012) for command line interfaces.”

We also added the “external resources” on the app as follows. 

“External resources: MiCloud does not take raw sequence data. For the raw sequence data processing and microbiome profiling, we recommend other popular and well-established bioinformatic pipelines, such as 

Nephele (https://nephele.niaid.nih.gov), Qiita (https://qiita.ucsd.edu), QIIME2 (q2studio) (https://qiime2.org) and PUMAA (https://sites.google.com/g.ucla.edu/pumaa) for web platforms, or 

QIIME (http://qiime.org), QIIME2 (q2cli) (https://qiime2.org), MG-RAST (https://www.mg-rast.org), Mothur (https://mothur.org), MEGAN (http://ab.inf.uni-tuebingen.de/software/megan6) and MetaPhlAn (https://huttenhower.sph.harvard.edu/metaphlan) for command line interfaces.”

Editor's Comments:

My first comment concerns the validation of the findings provided by MiCloud. You have used a dataset of a twin study to show what your pipeline is able to do and you presented the obtained data. But, there is no validation of this data for example by comparing these findings with those obtained on the same dataset by using another analysis tool.

(Response) We do not propose any new statistical methods in this paper. MiCloud is a web platform that makes user-friendly implementation of existing methods that have been only available through command line interfaces. Of course, different methods provide different results, but they do not rigorously tell which method is valid or not. We referenced the original articles for more details (e.g., theories, methodologies and validity issues). However, as you and the reviewer suggested, we described the rationale for the selected statistical methods in the Materials and Methods as follows.

“There are many other statistical methods that can be considered for microbiome downstream data analysis, but the rationale for the selected statistical methods (Fig 1) is in their popularity, statistical validity and easy interpretation/presentation of the results as follows. 

First, for the cross-sectional studies, statistical methods based on the independence assumption have been widely used. The Welch t-test and Wilcoxon rank-sum test (Mann and Whitney, 1947) can be used for non-covariate adjusted comparative analysis with a nice graphical presentation using box plots and summary statistics such as, mean, minimum, Q1, median, Q3 and maximum values. The linear regression, logistic regression, negative binomial regression, and beta regression models can be used for the continuous, binary, count and proportional response variables, respectively, with or without covariate adjustment, where the estimated regression coefficients, standard errors, confidence intervals, and P-values serve as a breadth of statistical inference facilities for the effect direction and size, variability and significance. The forest plot, line graph, and/or dendrogram can also efficiently summarize the results. Lastly, microbiome regression-based kernel association test (MiRKAT) (Zhao et al., 2015, Wilson et al., 2021) has recently been highlighted for the beta-diversity analysis with or without covariate adjustment, where the principal coordinate analysis (PCoA) plot (Torgerson, 1952) can nicely summarize the results.

Second, in the longitudinal/family-based microbiome data, the repeated measurements from the same subject or the subjects from the same family tend to be correlated with each other because of the shared genetic components and environmental factors (e.g., diet, residence, etc). Hence, the statistical methods based on the independence assumption described above are not statistically valid, leading to inflated type I error rates, for longitudinal/family-based studies. Hence, we selected a series of statistical methods that are based on the random effects models (i.e., the linear mixed model (LMM) (Laird and Ware, 1982) and generalized linear mixed model (GLMM) (Breslow and Clayton, 1993)) or generalized estimating equations (GEE) (Liang and Zeger, 1986) for both ecological and taxonomical analyses because of their well-known statistical validity (i.e., robust controls of type I error rate) for correlated data analysis. The results can also be presented using a breadth of statistical inference facilities, summary statistics and visualizations. 

More details on each sub-component are addressed in following sections.”

My second comment concerns the discussion, which is very short. Knowing that MiCloud seems to perform a restricted number (but very useful) analyses that cannot be handled by other existing web platform, I ask me for example why you have chosen to develop a new platform and not to improve one of these existing platform? More globally, I ask you to discuss more in deep, the advantages and the limits of your tool.

(Response) MiCloud handles more types of microbiome data analysis (e.g., covariate-adjusted analysis, longitudinal/family-based data analysis) that other existing web platforms, yet MiCloud cannot handle all possible types of microbiome data analysis. We revised the Discussion as follows.

“MiCloud extends web-based microbiome data analytics to covariate-adjusted analysis and longitudinal/family-based data analysis, yet MiCloud does not handle upstream data processing (raw sequence data processing and microbiome profiling) and all possible types of downstream data analysis.” 

We also included some description on cross-sectional vs. longitudinal methods in Discussion as follows. 

“Especially, in the longitudinal/family-based microbiome data, the repeated measurements from the same subject or the subjects from the same family tend to be correlated with each other due to the shared genetic components and environmental factors. Hence, the statistical methods based on the independence assumption, used in other existing web platforms or used for cross-sectional studies in MiCloud, are not statistically valid, leading to inflated type I error rates, for longitudinal/family-based studies. However, MiCloud additionally employs a series of statistical methods that are based on the random effects models (Laird and Ware, 1982) and GEE (Liang and Zeger, 1986) (Table 1) for both ecological and taxonomical analyses; as such, users can easily handle correlated data from longitudinal/family-based microbiome studies on our user-friendly web environments.”

We do not have access to other existing web platforms and thus cannot directly improve them. We used R Shiny to develop MiCloud because many of the current statistical methods and visualization approaches are freely available through R libraries. We included some description in Discussion as follows. 

“We used R Shiny to develop MiCloud. Many of the current statistical methods and visualization approaches are written in R language, and they are freely available through R libraries; hence, we could easily transfer them to MiCloud. Galaxy (Afgan et al., 2016) is another popular platform to develop web applications in computational biology, but many of the current Galaxy applications are written in different programming languages, and focus more on raw sequence data processing and genome/microbiome profiling, rather than downstream data analysis. It is beyond the scope of this research to compare R Shiny to Galaxy, but we would say that R Shiny can be better for downstream data analysis, while Galaxy can be better for upstream data processing.“

---

## [Editor Report · Decision Letter 1]

19 Jul 2022

MiCloud: A unified web platform for comprehensive microbiome data analysis

PONE-D-22-09539R1

Dear Dr. Koh,

We’re pleased to inform you that your manuscript has been judged scientifically suitable for publication and will be formally accepted for publication once it meets all outstanding technical requirements.

Kind regards,

Jean-François Humbert

Academic Editor

PLOS ONE
---

## [Editor Report · Acceptance letter]

22 Jul 2022

PONE-D-22-09539R1 

MiCloud: A unified web platform for comprehensive microbiome data analysis 

Dear Dr. Koh:

I'm pleased to inform you that your manuscript has been deemed suitable for publication in PLOS ONE. Congratulations! Your manuscript is now with our production department. 

Kind regards, 

on behalf of

Dr. Jean-François Humbert 

Academic Editor

PLOS ONE